# Magnetic-Responsive Bendable Nozzles for Open Surface Droplet Manipulation

**DOI:** 10.3390/polym11111792

**Published:** 2019-11-01

**Authors:** Lizbeth O. Prieto-López, Jiajia Xu, Jiaxi Cui

**Affiliations:** 1INM-Leibniz Institute for New Materials, Campus D2 2, 66123 Saarbrücken, Germany; 2Abifor AG, Untere Mühlewiesen 8, 79793 Wutoschingen, Germany; jiajia.xu@abifor.com

**Keywords:** droplet manipulation, magnetic responsive elastomers, open-surface microfluidics

## Abstract

The handling of droplets in a controlled manner is essential to numerous technological and scientific applications. In this work, we present a new open-surface platform for droplet manipulation based on an array of bendable nozzles that are dynamically controlled by a magnetic field. The actuation of these nozzles is possible thanks to the magnetically responsive elastomeric composite which forms the tips of the nozzles; this is fabricated with Fe_3_O_4_ microparticles embedded in a polydimethylsiloxane matrix. The transport, mixing, and splitting of droplets can be controlled by bringing together and separating the tips of these nozzles under the action of a magnet. Additionally, the characteristic configuration for droplet mixing in this platform harnesses the kinetic energy from the feeding streams; this provided a remarkable reduction of 80% in the mixing time between drops of liquids about eight times more viscous than water, i.e., 6.5 mPa/s, when compared against the mixing between sessile drops of the same fluids.

## 1. Introduction

Droplet manipulation is the core to many technological and biomedical applications spanning from inkjet printing [1,2,3] and additive manufacture [4,5] to drug development/screening [6,7] diagnostics [8,9,10] and point of care devices [11,12]. There is a growing interest in developing open-surface systems that can manipulate small volumes of liquids in the form of drops as an alternative to the limitations imposed by the closed physical conditions in standard microfluidic chips. Open-surface systems can enable the independent and simultaneous progress of intricate chemistries while offering direct access to effluents for observation and analysis at any time or stage in the process. Some of the common complications faced in standard microfluidic systems such as microchannel clogging, cross-contamination and axial-dispersion of the sample can be circumvented in these open-surface systems.

Recently, several open-surface microfluidic systems have been developed to manipulate individual drops of fluids in a precise manner by dispensing them on specialized substrates; most of these systems rely on electrowetting [13,14,15,16], magnetic fields [17,18,19,20], or surface acoustic waves [21,22] to drive the droplet motion. They are often called “digital-microfluidics” because of the particular operation of the discrete volumes of liquid. These systems usually require expensive laborious fabrication and additional components to provide the source of energy for the droplet motion. As an alternative to these expensive devices, a class of low-cost systems has also emerged [23,24] which relies on the effects derived from surface tension. These systems are based on the use of wettability maps to confine and transport drops from one section to another on a substrate. For example, capillarity developed in the porous matrix of paper has been applied to run the “pump-less” motion of fluids in paper-based microfluidics [10,25,26,27]. Also Laplace pressure has been used to drive the motion of fluids through special designs of wettable patterns, such as wedge-shapes [28,29] or dots connected through a narrow line [30], on superhydrophobic substrates. Moreover, wettability gradients [31,32,33] and surface deformation [34,35] have also been explored as means to provide unidirectional liquid transport. Although the basic droplet operations such as transport and mixing have been successfully demonstrated in these systems, to the best of our knowledge, it is still challenging to achieve complex fluid manipulation such as the re-programmability of the fluidic routes.

Magnetic particles, typically of Fe_3_O_4_, with micrometric dimensions embedded in elastomeric matrices have been widely used for fast, easy, and reversible actuation of structures in a number of applications. Elongation [36], contraction, coiling [37], and deflection [38] of structures, as well as the tuning of the elastic modulus of elastomers [39] are some examples of the capabilities of such magnetic composites.

Herein, we describe a new platform for droplet handling which can trigger the mixing between open-to-air drops; it enables versatility in the combinatorial paths, facilitates further processing of the fluids, and allows easy observation and direct access to the products for analytics. It is based on an open array of bendable nozzles that are dynamically controlled by a magnetic field. The nozzles have a hybrid structure constituted by a main body made of commercial polydimethylsiloxane (PDMS) and a magnetic responsive tip made of a magnetic composite based on PDMS and Fe_3_O_4_ magnetic particles. This structure was fabricated with a two-step soft lithography process. By bringing together the tips of these magnetic responsive nozzles we can mix, split, or transport the drops pumped in/out of them. This system offers an advantage for droplet mixing over simple droplet coalescence by harnessing the fluid velocity from the nozzles to promote internal advection. We demonstrate the potential of our platform to control complex combinatorial routes between droplets by showing a two-step reaction, a dilution sequence, and a splitting and re-mixing examples.

## 2. Materials and Methods

### 2.1. Droplet Platform

A sketch of the three-part mold specially designed to fabricate our platform is shown in Figure 1a. The main body of our mixing platform was made of Sylgard 184 with a base to catalyst mix ratio of 10:1. The bendable nozzles were made of a magnetic composite prepared with Sylgard 184 base-to-catalyst ratio 10:0.5 combined with a 40% in weight of Fe_3_O_4_ microparticles of average diameter ~5 µm (Neotexx X003-050, Neomagnete, Berlin, Germany). The bending control was provided by a magnetic field from a 1 mm × 1 mm neodymium magnet of maximum energy product of BH_max_ 342–358 kJ/m^3^ (Magnet-shop SM-01x01-N, grade N45, pull force ~ 25 g, Magnets4you GmbH, Lohr a. Main, Germany).

### 2.2. Liquid Properties

The contact angle between the different fluids and composites was measured with a contact angle measuring instrument (OCA 25 DataPhysics Instruments GmbH, Filderstadt, Germany). For these measurements thick films of the different materials, PDMS or composites, were prepared with flat and smooth surface. The viscosity of the test fluids was measured in a Rheometer Physica MCR 300 (Anton Paar, Ostfildern-Scharnhausen, Germany) at 20 °C using 11 mL of solution. The surface tension was measured with a tensiometer (Wilhelmy, Krüss K12, Hamburg, Germany) with a platinum plate method.

### 2.3. Optical Setup for Mixing Images

The tracking of the mixing process was done by laser-induced fluorescence (LIF) and image processing. With the aid of a neutral gray filter (1% transmittance) and a cylindrical lens, the emission from a solid state laser (CrystaLaser GCL-025-S, CLASS IIIb, λ = 520–540 nm, maximum power 500 mW) was converted into a light sheet of about 100 µm width. This light sheet excited the fluorescence from the droplets of diameter between 1.5 to 2 mm (~2 µL) containing Rhodamine B which absorbs at ~543 nm and emits at ~565 nm. An orange color filter glass type OG550 was placed in front of the USB camera (DinoLite AM4113ZT, Dino-Lite Europe, Naarden, Netherlands) to remove the excitation light reflected on the surface of the droplet; this allowed a clearer observation of the fluorescence inside the droplet excited by the penetrating laser. The images were recorded at a frame rate of 30 fps with a resolution of 1.3 megapixels.

Two pairs of liquids were used in these fluorescence tracking experiments. Pair 1, with low viscosity of η = 0.8 mPa/s, involved pure MilliQ water as the non-fluorescent fluid and a solution of 0.05 mg/mL Rhodamine B (≥95%, Sigma-Aldrich R6626, Sigma-Aldrich Chemie GmbH, Taufkirchen, Germany) as the fluorescent. Pair 2, with higher viscosity of η = 6.5 mPa/s, was formed by a solution 50% by volume of glycerol (99+%, Alfa Aesar A16205, Thermo Fisher GmbH-Alfa Aesar, Karlsruhe, Germany) and water as the non-fluorescent fluid and the same solution with 0.085 mg/mL Rhodamine as the fluorescent counterpart. In both pairs the surface tension difference was low; (σ_1_ − σ_2_)/σ_1_ ≤ 0.1. Liquid pair 3 was formed by MilliQ water and an ethanol–water solution 50% with food colorant as the counter-part. This pair was used to exemplify the effect of large surface tension difference, (σ_1_ − σ_2_)/σ_1_ = 0.6, between the mixing fluids. This mixture was only optically tracked and no mixing index was computed from these experiments. The liquids were dispensed with a syringe pump (Kd Scientific, Legato 101, KD Scientific Inc., Holliston, MA, USA) at an effective flowrate of 50 µL/min.

### 2.4. Mixing Quantification

To track the evolution of the mixing process we computed a mixing index, MI, based on the average and the standard deviation of the gray-value of the gray-scale-images from LIF. Although the standard deviation alone can be used to quantify the homogeneity of the pixel brightness within a region, the average-gray-value is used to account for the variations in the illumination during the capture of a frame. We defined the following mixing index:MI = 1 − (σ_i_/μ_i_ − σ_∞_/μ_∞_)/(σ_0_/μ_0_ − σ_∞_/μ_∞_)(1)
where σ is the pixel gray-value standard deviation and µ is the average pixel gray-value. These values were evaluated considering the entire droplet region. The sub-indices 0, I, and ∞ refer to the initial, current, and final frame (after ~15 min) respectively. So the MI values grow from 0 for the initial not-mixed state to 1 for a fully mixed state. The image processing was carried out with Fiji, ImageJ [40].

## 3. Results and Discussion

### 3.1. The Platform for Droplet Manipulation

Figure 1b shows the design of our platform. It consists of a top layer with the array of bendable nozzles supported on an interfacing layer with embedded channels and ports. The special three-part mold designed to fabricate the array of bendable nozzles is shown in Figure 1a. This consists of a bottom plate which supports tungsten carbide pins of 300 µm diameter; these pins create the internal channel of the nozzles, a central plate which shapes the body of the nozzles, and a top plate which provides a flat counterface and the alignment for the pins. An aspect ratio of L/D = 5 was chosen for the nozzles (Figure 1c) to render enough flexibility in bending. The geometrical parameters of the template shown in Figure 1c were based on two constraints: the diameter of the feature molding the internal channel of the nozzles, in this case the tungsten carbide pins, and the space required to accommodate the actuating magnets of 1 mm in diameter. The limiting factor in the fabrication was the structure molding the internal channel of the nozzles; this structure must have an aspect ratio larger than L/D = 5 in order to mold the internal channel and connect it through the base to the main reservoir. In our case the molding pins had an aspect ratio of L/D = 8.

The complete fabrication process is sketched in Figure 1d. The nozzle array layer was prepared in two steps, first the uncrosslinked magnetic composite was poured into the mold, degassed, and partially crosslinked; and then the clear PDMS was added to fill-up the rest of the mold. The backing layer with the microchannel network was integrated at the bottom of the nozzle array to provide the inlet/outlet of the fluids through the nozzles. This backing layer was fabricated with standard techniques in microfluidics. The integration of both layers was achieved by plasma-assisted bonding.

Due to the simple fabrication based on molding steps followed by the bonding of both constituting parts, in principle, bigger templates and larger arrays could be easily produced once the appropriate molds are fabricated. However, a different fabrication approach, such as 3-D printing or two-photon lithography, would be required to scale it down; this is due to the structural requirements of the feature molding the internal channel in the nozzles which becomes more critical at smaller scales and so does the correct alignment between different features in the bonded parts which increases the difficulty for the assembly.

The magnetic composite forming the nozzles was prepared with a mixture of PDMS precursor and Fe_3_O_4_ microparticles. It is known that the addition of microparticles or fillers increases the stiffness of elastomers. Therefore to retain the flexibility of the nozzles, at the fixed geometry, we reduced the cross-linked density of the PDMS matrix. Decreasing the cross-linker concentration to 4.8% allowed us to keep the Young’s modulus of the composites at the desired low value despite the addition of microparticles (Appendix A). A content of 40% in weight of magnetic particles showed the best performance in our application; we evaluated this based on the ratio of the response-distance/magnetic-mass within the content range between 40% and 60%. This ratio indicates the maximum distance at which the nozzle can respond to the controlling magnetic field, and snap into contact, over the mass of magnetic particles in the composite (details in Appendix A). Although this ratio is more relevant for mass production, viz., to economize on reagents, the limiting factor is actually the response-distance; this determines the maximum distance at which the nozzles can be actuated. Pillars located at longer distances from the magnet do not come into contact. The response-distance for a magnetic-mass ratio of 40% was of ca. 22 µm/g, 19 µm/g for 50%, and 17 µm/g for 60%. A minimum content of magnetic particles of 40% was needed to meet the response-distance required by our geometrical constraints; higher content of magnetic particles in the composite did not improve significantly the response. We also evaluated the effect of particle arrangement on the nozzle bending response. We introduced a preferential arrangement of the magnetic particles relative to the principal axis of the nozzles by curing the magnetic composite under the action of an external magnetic field. Three configurations were evaluated: ‘parallel’ with the magnetic particles arranged along field lines parallel to the principal axis of the nozzles, ‘perpendicular’ with magnetic particles arranged along field lines perpendicular to the principal axils of the nozzles, and ‘random’ with a random distribution and orientation of the magnetic particles, cured without external field. The results are summarized in Table 1 (more details in Appendix A). We observed that for a determined crosslinker concentration and determined magnetic particle content in the composite, the longest distance between magnet and nozzle for sudden contact varies with the orientation/distribution of the magnetic particles within the nozzle, relative to its axis. Nozzles with randomly oriented and homogeneously distributed particles exhibited the largest distance at which the sudden snap into contact with the magnet occurs. These nozzles showed, for different orientations of the actuating magnet, a distance 29% (poles), 40% (vertical), and 60% (horizontal) superior to the distance observed in the nozzles with particles arranged along field lines parallel to the principal axis; which represented the worst case (Table 1). The higher stiffness in these nozzles is due to the structural reinforcing provided by the chain of particles arranged along the principal axis which render a higher opposition against bending. This effect has been documented before [38,39,41].

### 3.2. Properties of the Liquids Handled in the Experiments

The contact angle measured against PDMS and composite, the surface tension and the viscosity of the various liquids used in this work are summarized in Table 2. The rather similar contact angle of the liquids observed against PDMS and composite surfaces suggests that a similar equilibrium gas-liquid-solid exists between the fluids and both types of substrates. This is due to the complete encapsulation of the magnetic particles by PDMS in the composite, which renders an external PDMS surface as demonstrated by EDX analysis (Appendix A). The slight change in the contact angle can be related to the slight increase in surface roughness with the introduction of particles as observed by interferometry (see Appendix A).

Therefore, given that the material in actual contact with the fluids is PDMS, the solvent compatibility of our template corresponds to that of PDMS. Nevertheless, long exposures to fluids that mildly swell PDMS could reach the magnetic particles after some time and interact with them. In this study, the templates have been continuously exposed for more than three months to ethanol, glycerol, and water mixtures without integrity degradation of the template.

### 3.3. Droplet Stability on Nozzles

We first tested the capability of a nozzle to support a droplet. With the current nozzle geometry, i.e., external diameter of 800 µm, a drop of colored water (γ ~ γ_water_ = 72.3 mN/m) of about 5 µL could remain stable on a nozzle with a nearly perfect spherical shape; Figure 2a. Further increase in the volume induces a large droplet deformation because of gravity. The enlargement of the nozzle tip surface with the introduction of a “mushroom-like” tip expanded the operational volume up to 8 µL on individual nozzles; Figure 2b. In the case of liquids with lower surface tension than water, as in the case of the solution ethanol-water 50% (γ = 28.8 mN/m), the maximum operational droplet volume was inferior; about 3 µL with normal tips and ca. 5 µL with the extended tip. To ensure the stability of drops of all fluids throughout our experiments and to maintain a spherical droplet contour, we used 2-µL drops on individual nozzles.

### 3.4. Basic Droplet Operation: Mixing Using the Template

The coalescence and mixing of droplets are tasks not so easily achieved because of the intrinsic difficulties imposed by the large surface-to-volume ratio and the developed laminar flows in the micro-level. In this platform, the fluid velocity from the fresh streams injected into the growing drop is harnessed to boost the internal advection. The greater kinetic energy introduced in the system by the injected flowrates leads to a faster mixing in comparison with coalesced sessile drops. To demonstrate this, we set two different mixing configurations diagrammatically shown in Figure 3 which we labelled as sessile-mix and enhanced-mix. In the sessile-mix configuration (Figure 3a), a drop of “pure” fluid is formed on the tip of two nozzles and they are brought together to coalescence 20 s after the complete release of the set volume of 2 µL. The droplet impacting speed in these experiments was about 30 µm/s (from image sequences), which is well below the impacting speed of 1 m/s reported to affect the mixing rate [42,43,44]. In the enhanced-mix configuration (Figure 3b), the tips of the nozzles are brought together into a contact position before the fluids-to-mix are released. In this case, tiny drops of “pure” fluids coalesce at the beginning of the dispensing process so the mixing starts simultaneously with the dispensing. A faster mixing is expected at higher flowrates because of a greater kinetic energy.

Figure 3 shows a typical LIF grayscale-image sequence obtained from the experiments with liquid-pairs 1 and 2. The sequences in (a) correspond to experiments in sessile-mix configuration and those in (b) to the enhanced-mix. In these images, the clear boundary between “pure” fluids remains only in the early stage of the mixing; the fluorescent liquid corresponds to the bright region and the non-fluorescent liquid to the dark region. The progress of the mixing is indicated by the gradual improvement in brightness homogeneity within the droplet region. The reflections from the drop surface hindered the clarity of the process, nevertheless these images provided a reasonable semi-quantitative way to evaluate the progress of the mixing [43,44,45]. The color-image sequences in Figure 3 correspond to experiments with liquid-pair 3; these are used to demonstrate the further improvement in the mixing due to Marangoni flow promoted by the large difference in surface tension between fluids.

The mixing indices (MI), computed with Equation (1) and the gray-value from the image sequences, are plotted against time in Figure 4. The plots in Figure 4a,b show the averaged (more than four tests each) MI obtained for liquid-pair 1 and pair 2, respectively, at an actual dispensing flowrate of 50 µL/min. Different nominal flowrates were evaluated; however, the actual delivered flowrate was similar for all settings and corresponded to 50 µL/min. This demonstration is included in the Appendix A. The squares in Figure 4 show the semi-experimental data and the lines represent the best fit of MI = A + Be^Ct^ to each data set. In liquid-pair 1 (Figure 4a), the reduction in the mixing time using the enhanced-mix mode was of nearly 40%; from ca. 80 s with the sessile-mix down to about 50 s. This is comparable to the mixing based on electrowetting, which was reported of about 20 s for 1.75-µL droplets [46], or on wettability gradients [47], reported at about 100 s for complete mixing of 0.3-µL droplets.

A very marked reduction of more than 80% was observed in the mixing of liquid-pair 2 (Figure 4b), from nearly 8 min down to about 80 s with the enhanced-mix configuration. The higher viscosity reduces the mobility of the species and increases the dissipation rate of the mixing driving forces, which results in longer mixing times. The delay is more evident in the sessile-mix mode where the mixing driving force is limited, mainly arising from the excess surface energy after coalescence [48] and from the marginal surface tension difference [49]. The streams from the nozzles in the enhanced-mix mode, promote the mobility of the liquids within the drop during and after formation (see Appendix A), thus enabling the faster mixing observed in Figure 3 and Figure 4. This highlights the advantage of harnessing the feeding streams to induce internal flow to promote the faster mixing in liquids with higher viscosity than water. Additional chaotic advection in the combined drop can be triggered by a gradient in a fluid property such as density, temperature, viscosity, surface tension [49,50], or by dissimilar drop size [48]. We introduced in our experiments a surface tension gradient to induce the Marangoni flow. This was carried out with liquid-pair 3, whose large surface tension difference provided the gradient; these results are the color image-sequences in Figure 3. A significantly faster mixing was observed in both configurations because of the chaotic advection promoted by the Marangoni flow. However, once again, a very marked reduction of the mixing time was observed in the enhanced-mix configuration, from 50 s in the sessile-mix to less than 0.33 s with the enhanced-mix.

### 3.5. Versatility for Combinatorial Paths

The possible combinatorial paths in this type of platform are determined by the number of nozzles in the array and their layout. The use of magnetic fields to activate the bending entails that all nozzles within the working range of the magnetic field will simultaneously come into contact. Therefore, the array layout must be designed to ensure that only the first neighbors to the magnet are actuated; this means only three nozzles in a hexagonal array, Figure 5a, or four nozzles in a square array, Figure 5b, and so on. The layout also determines the maximum number of fluids that can be combined in a single operation, for example, in the hexagonal-array three nozzles can come into contact; two nozzles can provide different fluids for reaction and the third nozzle can be used to transport the product into the next stage of the process. Here we demonstrate the potential application of a platform with a hexagonal-array through an example of a two-step process, a dilution sequence and a splitting-recombination process.

#### 3.5.1. Two-Step Process

Two color changing reactions were triggered consecutively in this example. As the first step we use the changing color reaction between phenolphthalein and sodium hydroxide, which results in a bright magenta color. The second step involved the neutralization of this product with the addition of citric acid; which results in the disappearance of the color. Figure 5c shows the sketch of the path followed to completion of this two-steps process. The step-by-step sequence is shown in Figure 5d–k. Initially the magnet is at the “off-range” or down position and the nozzles remain in their upright configuration (Figure 5d). When the magnet is raised into the “in-range” or up position, the three closest nozzles snap into contact (Figure 5e). Two nozzles supply the two reagents, 2 µL of NaOH 50 mM solution and 2 µL of phenolphthalein 0.5 g/L in ethanol–water 50% solution, the third nozzle is used to transport the product to the following stage. The product from this reaction has a bright magenta color (Figure 5f) which is the characteristic indication for a basic solution. The mixing time in this step was about 0.3 s for a surface tension difference of (σ_1_ − σ_2_)/σ_1_ = 0.6; which agrees with the observations when introducing Marangoni flow. Video 1 (Appendix A) shows the nearly instantaneous mixing with the dispensing of the drops. Next, the magenta product is split into three parts; the larger portion is sucked-in by the transporting nozzle and the remaining volume can stay on the tips of the nozzles because of the surface tension and re-shape into smaller drops after the separation of the tips (Figure 5g,h). Then a second magnet is placed at the up position (Figure 5h) to enable the second part of the process. Here the transported portion (~1.5 µL) of the magenta product combines with 2 µL of citric acid 50 mM solution (Figure 5i). In this second stage, the mixing takes about 3 s to completion and it is perceived in the clear color of the new product drop (Figure 5j). In this case, the surface tension difference between fluids is reduced to (σ_1_ − σ_2_)/σ_1_ = 0.46, thus the weaker effect of Marangoni flow and the slower mixing. As indicated by the arrows in Figure 5c, the transfer of this product could follow on to different stages for additional steps such as splitting, mixing, or reaction with another fluid, or analysis and so on.

#### 3.5.2. Dilution Sequence

The consecutive dilution of solutions is a common practice in most laboratories. Here we exemplified the potential application of this template to achieve certain dilution of an analyte. To illustrate the dilution we used a blue food colorant in an ethanol solution 50% as the initial solution (see Table 2). Figure 6 shows the process step by step. Initially, (Figure 6a) 2-µL drop of the starting blue solution is (Figure 6b) combined with 2 µL of fresh water, this corresponds to dilution 1. The product from dilution 1 is then (Figure 6c) sucked-in by the third nozzle. The position of the magnets is rearranged (Figure 6d) to bring the next set of nozzles into contact, and then (Figure 6e) a ~2-µL drop of dilution 1 is (Figure 6f) mixed with ~2 µL of fresh water. This corresponds to dilution 2. The product is then transferred (Figure 6g,h) to the next stage where (Figure 6i) a ~2 µL drop of this dilution is (Figure 6j) mixed once again with ~2 µL of fresh water for dilution 3. This product is then transferred (Figure 6k,l) to the last dilution stage where it combines once more with (Figure 6m,n) fresh water. The combinatorial route followed in this process is sketched in Figure 6o. In this example, because of instrumental limitations, most of the pumping-in and -out has been carried out manually; therefore the lack of precision in the released volumes.

#### 3.5.3. Splitting-Recombination

The division and distribution of reagents for subsequent processing is also possible in this template. Figure 7 illustrates the splitting of a drop of ~6 µL of a yellow solution into two drops of ~3 μL for its further release and simultaneous mixing with a red and a blue solution in a second and a third stage, respectively. The three starting solutions: red, yellow, and blue, share similar concentration of food colorant (0.2 g/mL) in a 50% ethanol solution; therefore, one can safely assume that the properties of these liquids are rather similar (see Table 2). This process illustrated in Figure 7 starts by bringing the set of nozzles at the bottom center of the template into contact (Figure 7a), then follows the release of the yellow solution (Figure 7b) through the bottom center nozzle. In this example, the splitting of the drop is carried out by the (Figure 7c) sucking action of the two central nozzles in the middle row; this ensures the controlled splitting of the relatively large volume. The splitting of large volume drops (in this case larger than 2 µL) by separation of the nozzles resulted in drop instability and fall out. After the rearrangement of magnets (Figure 7c,d), the split volumes from the initial yellow drop are released in two different stages (Figure 7e), then 2 μL of red and blue solutions are simultaneously injected into the yellow drops. Once again, the homogeneous mix (Figure 7f) can be observed in less than 2 s (see Video 2 in Appendix A). Figure 7g illustrates the flow path followed in this process, 2-μL drops of the initial red, yellow, and blue solutions are shown as reference.

As observed throughout the examples here illustrated, the surface tension of fluids has a very important role in this template. On one side, it is essential for keeping the droplets on the pillars and also to speed up the mixing when a surface tension gradient develops inside them. However, the inherent residual fluid on the tip of the nozzles after each interaction, could become an issue for reactions and processes with critical restrictions. We envisaged further processing of the nozzle tip, such as hydrophobization, to prevent the residual liquids and diminish this risk of cross-contamination.

Another major issue in the applicability of the template concerns the pumping of fluids. The precise control of the released and sucked volumes is essential for most applications; therefore, the integration of a reliable pumping system in our template is of primal need in order to complete the complex process. Furthermore, the automatization and remote control of the magnetic fields triggering the bending of the nozzles could add a great advantage by reducing the activation time required to set the different nozzles stages.

## 4. Conclusions

We have developed a novel platform for droplet manipulation in an open-surface manner. This system consists of an array of bendable nozzles whose tips can be dynamically brought into contact to enable the transport, mixing, and splitting of droplets. We have demonstrated that by harnessing the kinetic energy from the fresh fluid stream feeding the drops, the mixing time can be drastically reduced in more than 80% between liquids with viscosity eight times that of water; this is from 8 min in sessile-mix down to 80 s in enhanced-mix. This reduction is less with liquids of viscosity similar to water; from 80 s to 50 s between 2-µL drops. These reductions were obtained at the relatively low flowrate of 50 µL/min. When Marangoni flow was introduced in the mixing, the time for complete homogenization was further reduced to about 0.33 s between 2-µL drops of fluids with rather similar viscosity to water. We exemplify the versatility for potential combinatorial paths offered by this template through a two-reaction process, a dilution sequence, and a splitting-recombination process. We foresee the great interest for technological and biomedical applications that require viscosity operational range greater than that of water.

## Figures and Tables

**Figure 1 polymers-11-01792-f001:**
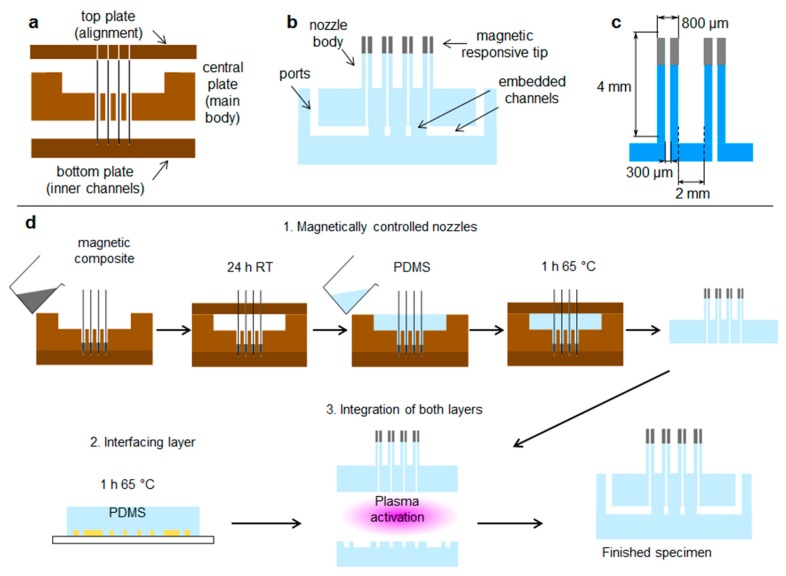
Design of platform for droplet manipulation. (**a**) Side view of constituting plates of our three-part mold. (**b**) Side view of the finished platform. (**c**) Nozzle aspect ratio and geometrical parameters. (**d**) Fabrication process.

**Figure 2 polymers-11-01792-f002:**
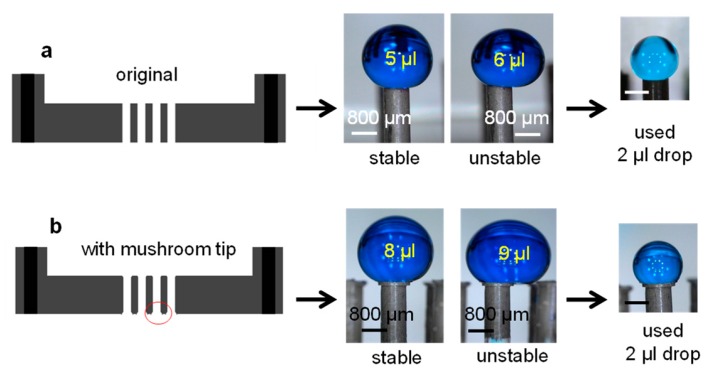
Extending the operational droplet volume. Central part of the mold used for fabrication of the nozzles and images of maximum droplet volumes supported by the tips (**a**) normal and with (**b**) extended mushroom-like tip.

**Figure 3 polymers-11-01792-f003:**
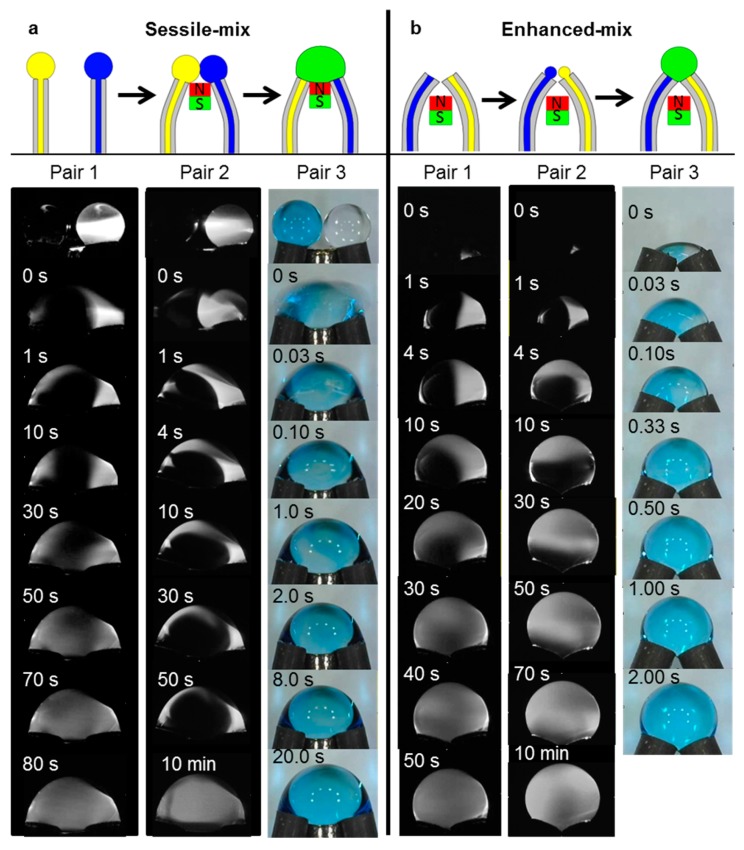
Mixing configurations. Laser-induced fluorescence (LIF) and color image sequences of (**a**) sessile-mix and (**b**) enhanced-mix of liquid pair 1: water and water with rhodamine, pair 2: water-glycerol and water-glycerol with rhodamine, and pair 3: water and water-ethanol with food colorant.

**Figure 4 polymers-11-01792-f004:**
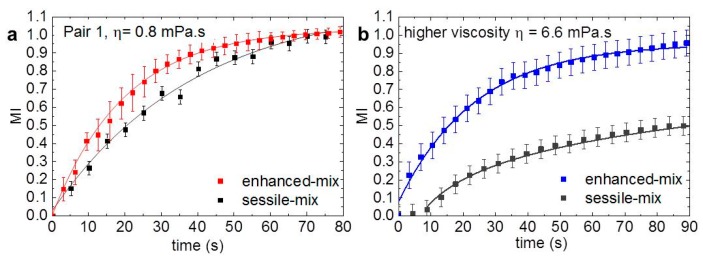
Mixing index (MI) change in time. Average curves of an actual flowrate of 50 µL/min for (**a**) pair 1 and (**b**) pair 2.

**Figure 5 polymers-11-01792-f005:**
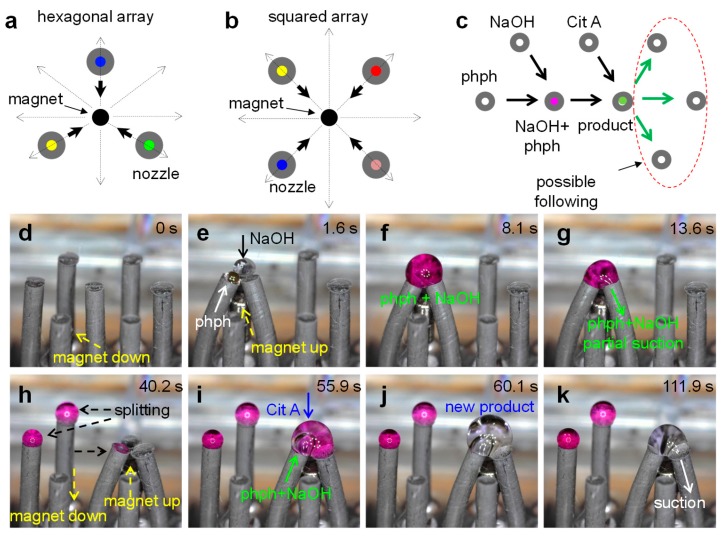
Versatility for combinatorial paths. Array of magnetic nozzles with a (**a**) hexagonal layout and a (**b**) square layout interacting with a magnetic field (top view). (**c**) Example of a two-step reaction path diagrammatically sketched. (**d**–**k**) Step-by-step of two-step reaction.

**Figure 6 polymers-11-01792-f006:**
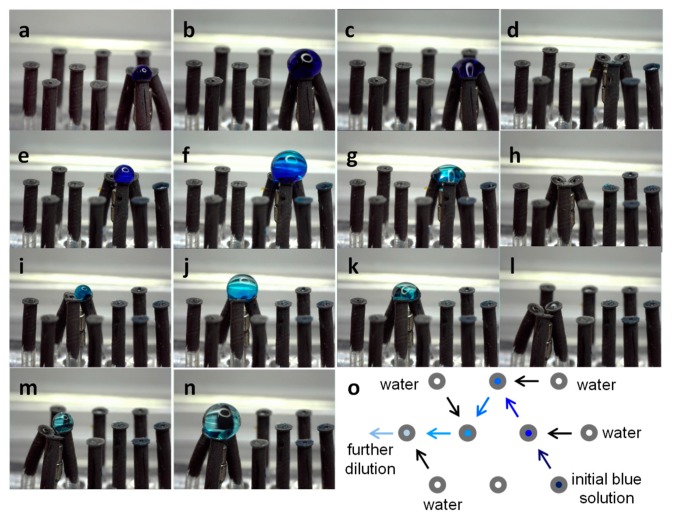
Dilution sequence. (**a**) Magnet #1 is raised to bring the three nozzles at the bottom-right edge of the template into contact, 2 µL of the initial blue solution (50% ethanol in water) are released. (**b**) Initial dilution with water stream. (**c**) Product from dilution 1 sucked-in for transfer to next stage. (**d**) Magnet #1 down and magnet #2 up, next set of nozzles are brought into contact. (**e**) Droplet from dilution 1 released and (**f**) mixed with fresh water for second dilution step. (**g**) Dilution 2 product sucked-in. (**h**) Magnet #2 down and magnet #3 up, next set of nozzles in contact. (**i**) Droplet from dilution 2 released and (**j**) mixed with fresh water for the third dilution. (**k**) Product from third dilution sucked-in for transfer. (**l**) Magnet #3 down and magnet #4 up, next group of nozzles into contact. (**m**) Drop from dilution 3 released and (**n**) mixed with fresh water for the fourth dilution. (**o**) Sketch of the route followed for this dilution sequence.

**Figure 7 polymers-11-01792-f007:**
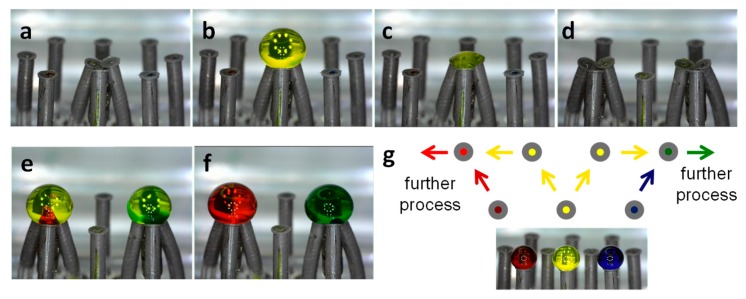
Splitting-recombination process. (**a**) First group of nozzles brought into contact. (**b**) Release of yellow solution and (**c**) split into the other two nozzles by suction. (**d**) Rearrangement of the magnets, two groups of nozzles ready for combination. (**e**) The two portions from the initial drop are released and then the mixing solutions red and blue, left and right, are simultaneously released. (**f**) The homogeneous mix is achieved in less than 2 s. (**g**) Sketch of the flows in this process.

**Table 1 polymers-11-01792-t001:** Magnetic particle distribution and their maximum distance for bending response.

Particle Distribution Relative to Pillar Axis	Testing Magnetic Field Alignment		
	Poles	Vertical	Horizontal
 random	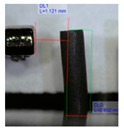 1110 ± 10 µm	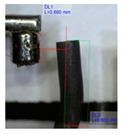 880 ± 10 µm	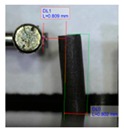 800 ± 10 µm
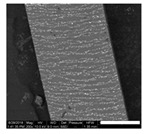 perpendicular	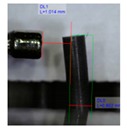 1000 ± 10 µm	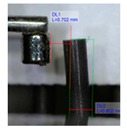 720 ± 10 µm	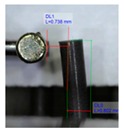 750 ± 10 µm
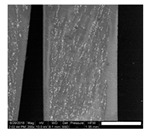 parallel	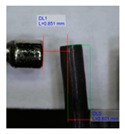 860 ± 10 µm	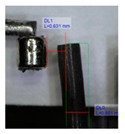 630 ± 10 µm	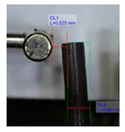 500 ± 10 µm

**Table 2 polymers-11-01792-t002:** Properties of liquids used in experiments.

Liquid	Contact Angle PDMS	Contact Angle Composite	Viscosity (mPa/s)	Surface Tension (mN/m)
Water	117° ± 1°	116° ± 2°	0.8	72.3
Water + Rhodamine B	111° ± 2°	115° ± 3°	0.8	65.5
Water + glycerol	115° ± 1°	116° ± 2°	6.6	56.4
Water + glycerol + Rhodamine B	113° ± 1°	114° ± 2°	6.4	63.0
Water + ethanol + color	67° ± 1°	69° ± 1°	2.5	28.8

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
