# Peer review of "Magnetic-Responsive Bendable Nozzles for Open Surface Droplet Manipulation"

_polymers, 2019, doi:10.3390/polym11111792_

Round 1

Reviewer 1 Report

This manuscript introduces the fabrication of magnetic-responsive bendable nozzles for open surface droplet manipulation. The designation and the ideas in this manuscript are very interesting and the obtained system has practical potential application. This work is detailed and specific. This manuscript should be accepted with minor revision. The detailed comments are as below:

The abstract of this manuscript is too concise to allow the readers to get all the meaningful information about this work. The abstract needs to be revised in more detail. In this work, the authors firstly fabricated a magnetically controlled nozzles with Fe3O4/PDMS composites. Based on this, the authors should introduce Fe3O4 more or less in the Introduction, including its basic properties and basic application fields. By doing this, this work can attract a wider readership. Some papers are suggested for the authors here: Journal of alloys and compounds, 2019, 775, 800-809; ACS sustainable chemistry & engineering, 2018, 6(11), 15598-15607; Nanomaterials, 2018, 8(6), 441; Analytical Chemistry, 2017, 89(24), 13472-13479. The authors should discuss the dispersion of Fe3O4 in polymers. I doubt if the aggregation of the magnetic particles will influence the performance of the system. The authors should consider the influence of the polarity of different sample solutions and the amount of Fe3O4 added on the wettability of the nozzles. Will this mean that the system is specific to a class of substances? The authors should state in their manuscript why they choose the displayed aspect ratios (4 mm, 300 um, 2mm and 800 um) of the nozzle in Figure 1. Is this due to the technical constraints that the relevant size cannot be changed, or is it due to other considerations? The authors should organize the picture information according to the order in which it appears in the manuscript. For example, the author refers to Figure 1b before referring to Figure 1a. This is not normal in scientific report.

Reviewer 2 Report

This manuscript described a system consisting of an array of magnetically-responsive bendable nozzles whose tips can be dynamically brought into contact to enable the transport, mixing and splitting of droplets. While the system itself is interesting, there are flaws in this work and more results should be presented to demonstrate the potential of the system. The following comments should be addressed before this manuscript can be accepted for publication.

In line 144, the authors stated that “for a determined crosslinker and magnetic particle concentration, the optimized response i.e. the longest distance between magnet and pillar for sudden pull-in, is achieved with a proper configuration of the magnetic particles”. What is the advantage of the longer distance between magnet and pillar? In line 166, the authors stated that they used 2ul drops on the nozzles. The image of a 2ul drop should be presented. In line 210, the authors attributed the lack of difference between the mixing rates in response to varying delivered flowrate to the inaccuracy of the flowrate, and claimed that the real flowrate was similar for all settings. If it is true, Figure 4a and b are meaningless and should be removed. Otherwise, readers may be misled by the results. A discussion on the scalability and practical application of the system is missing. To me, there was not sufficient evidence to demonstrate the potential of the system. Some suggestions for additional experiments include constructing a library of droplets containing contents at a gradient concentration by splitting and mixing colored droplets with water droplets in multiple steps. This would be useful to screen growth factors concentration in cell culture experiment for example. Another suggestion will be to perform multiplex assay using one sample input, first by splitting the sample into multiple small droplets, followed by mixing each droplet with different assay reagents for parallel reaction to occur.

Round 2

Reviewer 2 Report

The last comment was not addressed. I don't see a huge improvement in the quality of the manuscript.

Author Response

Two new sets of experiments have been completed attending the demands of the last comment; one set is to exemplify a series of dilutions of a blue solution, and the other is to exemplify the splitting of a volume and its followed mixing with different streams.

More supporting information has been added (and mentioned in the text) to support the general claims stated in the main text.

Efforts have also been devoted to improving thelanguage throughout the writing to make easier to the reader.